# Analysis of residual disease in periocular basal cell carcinoma following hedgehog pathway inhibition: Follow up to the VISORB trial

Shelby P. Unsworth[1], Christina F. Tingle[1], Curtis J. Heisel[1], Emily A. Eton[1], Christopher A. Andrews[1], May P. Chan[2,3,4], Scott C. Bresler[2,3,4], Alon Kahana[1,2,5]*

1 Department of Ophthalmology and Visual Sciences, Kellogg Eye Center, University of Michigan, Ann Arbor, MI, United States of America, 2 Rogel Comprehensive Cancer Center, University of Michigan, Ann Arbor, MI, United States of America, 3 Department of Pathology, University of Michigan, Ann Arbor, MI, United States of America, 4 Department of Dermatology, University of Michigan, Ann Arbor, MI, United States of America, 5 Department of Ophthalmology, Oakland University William Beaumont School of Medicine, Rochester, MI, United States of America

* alon.kahana@beaumont.org

**Data Availability Statement:** All relevant data are within the paper and its Supporting Information files.

## Abstract

Basal cell carcinoma (BCC) is a common skin cancer caused by deregulated hedgehog signaling. BCC is often curable surgically; however, for orbital and periocular BCCs (opBCC), surgical excision may put visual function at risk. Our recent clinical trial highlighted the utility of vismodegib for preserving visual organs in opBCC patients: 67% of patients displayed a complete response histologically. However, further analysis of excision samples uncovered keratin positive, hedgehog active (Gli1 positive), proliferative micro-tumors. Sequencing of pre-treatment tumors revealed resistance conferring mutations present at low frequency. In addition, one patient with a low-frequency SMO W535L mutation recurred two years post study despite no clinical evidence of residual disease. Sequencing of this recurrent tumor revealed an enrichment for the SMO W535L mutation, revealing that vismodegib treatment enriched for resistant cells undetectable by traditional histology. In the age of targeted therapies, linking molecular genetic analysis to prospective clinical trials may be necessary to provide mechanistic understanding of clinical outcomes.

**Trial Registration:** ClinicalTrials.gov Identifier: NCT02436408.

## Introduction

BCC is caused by deregulated hedgehog signaling, due to loss of function mutations in the receptor Patched1 (PTCH1) (up to 90%) or the downstream effector Smoothened (SMO) (~10%) [1,2]. Under normal signaling conditions, the receptor PTCH1 inhibits SMO in the absence of hedgehog ligand. However, upon ligand binding this inhibition is relieved and Gli transcription factors induce downstream target gene transcription. Vismodegib (Genentech Inc.) is a small-molecule SMO inhibitor approved by the United States Food and Drug Administration (FDA) to treat advanced and metastatic BCC [3]. Early trials reported clinical response in 30–58% of patients, with 0.6–46% of patients experiencing a complete response

**Funding:** AK: Research Grants from Genentech, Research to Prevent Blindness, Kellogg Eye Center, and The University of Michigan Rogel Comprehensive Cancer Center. The funders had no role in study design, data collection and analysis, decision to publish, or preparation of the manuscript.

**Competing interests:** AK was a consultant to Genentech, Inc., in 2018 and 2019 for the purpose of discussions with the Food and Drug Administration. AK has also had a consulting relationship with Stryker Corporation and BioTissue, Inc. over the past 12 months. SPU, CFT, CJH, ESE, CAA, MPC, and SCB have no relevant conflict of interest disclosures. This does not alter our adherence to PLOS ONE policies on sharing data and materials.

[4–8]. While vismodegib has a relatively high response rate, unfortunately a subset of these tumors become resistant and recur [9–11]. In a recent study of advanced BCC, 21% of patients who initially responded acquired resistance with a median regrowth time of 56 weeks [10]. Although many BCCs can be cured via excision [12], recurrence rates are high for advanced orbital and periocular BCC (opBCC), and excision may cause loss of visual function [13–18]. These data highlight two important concerns when considering vismodegib for advanced opBCC: 1) primary resistance, in which the treatment naïve tumor has a mutation downstream of PTCH1, or 2) secondary resistance, in which tumors acquire or select for mutations that circumvent SMO inhibition.

Case studies have highlighted the potential efficacy of vismodegib for preserving vision in patients with opBCC [19–24]. In 2015, we initiated the VISORB study: VISmodegib for ORbital and periocular Basal cell carcinoma [25]. Study subjects were prescribed 150 mg vismodegib daily (orally) for up to 12 months or until disease progression or unacceptable toxicity, upon which surgical excision was recommended. Patients were followed for up to 1 year after initiation of vismodegib. The final visit for patients who elected to undergo excision was 1 month (±1 month) following surgery. Baseline patient characteristics can be found in Table 1. The primary goal of the study was to preserve end-organ function while treating advanced opBCC. 100% of patients achieved a successful visual function outcome, as measured by a Visual Assessment Weighted Score (VAWS) [25]. 56% of patients achieved a complete clinical response to vismodegib, and of patients who elected to undergo excision post-treatment, 67% had no evidence of residual disease in excision samples (Table 2) [25]. Here, we further characterized pre- and post-vismodegib tissue samples from these patients to correlate clinical outcomes with gene expression and mutational analysis.

## Results

### Keratin positive micro-lesions are present in post-vismodegib surgical specimens

VISORB study dermatopathologists scored post-vismodegib surgical samples for presence of residual disease using standard hematoxylin and eosin (H&E) stained tissue sections. 27

**Table 1. VISORB study baseline patient characteristics [25].**

| | |
|---|---|
| **Number of Subjects** | 34 |
| **Age—year** | 67.1+/-12.2 |
| Median | 68.5 |
| Range | 48–95 |
| **Sex–number (%)** | |
| Male | 19 (56) |
| Female | 15 (44) |
| **Number of Tumors** | 35 |
| **Tumor Location–number (%)** | |
| Medial Canthus | 22 (63) |
| Lateral Canthus | 3 (8.5) |
| Lower Lid | 8 (23) |
| Brow/Orbit | 2 (5.5) |
| **Tumor Size—mm** | |
| Median | 21.5 |
| Range | 10–60 |

**Table 2. VISORB study outcomes [25].**

| Clinical Response (PE measurement) | # of subjects (%) 34 (100) |
|---|---|
| Complete Response | 19 (56) |
| Partial Response | 10 (29) |
| Stable Disease | 2 (6) |
| Progressive Disease | 0 (0) |
| Not Evaluable | 3 (9) |
| **Histological Response** | **27 (100)** |
| No sign of disease | 18 (67) |
| Disease present (clear margins) | 6 (22) |
| Disease present (extending to margins) | 3 (11) |

patients elected to undergo excision after vismodegib treatment. Of these 27, 18 showed a complete clinical response to treatment, 8 showed a partial response, and 1 had stable disease. 1 patient underwent excision prior to clinical evaluation of treatment due to poor tolerance, and 1 patient's histological tumor subtype precluded clinical physical measurements. Of the 27 patients who underwent excision, 18 (67%) showed no sign of disease, 6 (22%) had residual disease with clear margins, and 3 (11%) had residual disease extending to margins (Table 2, Fig 1A and 1B) [25]. To further characterize these samples, we performed Keratin 5 (K5) immunostaining. 100% of "no sign of disease" samples stained for K5 contained at least 1 non-follicular lesion (micro-lesion). On average, "no sign of disease" samples contained 10.3 lesions per section. "Disease present" samples contained an average of 71.4 (Table 3). While "no sign of disease" samples displayed a significant reduction in both the total combined area of micro-lesions and the number of distinct micro-lesions compared to "disease present" samples (Fig 1C–1E, Table 3), the average area of each micro-lesion was not significantly different between the two groups (Fig 1D). We looked at consecutive H&E and K5 stained sections from "no sign of disease" samples and found that these micro-lesions are visible in H&E sections but indistinguishable histologically from normal hair follicles (Fig 1E). A subset of these regions also display peripheral palisading, found in some follicle regions, but also a histological characteristic of BCC (Fig 1E).

## Residual micro-lesions express Gli1 and are proliferative

To identify whether these micro-lesions were residual BCC, we performed RNAscope™ *in situ* staining for the hedgehog target gene, Gli1. This technique not only allows visualization of Gli1 expression, but also quantification, as each dot indicates one molecule of Gli1 mRNA [26]. Previous studies in mice and humans have shown that vismodegib treatment is highly effective at blocking Gli1 and other hedgehog target gene expression in normal hair follicles and sensitive BCC tumors [27,28]. Because patients remain on vismodegib up until the day of surgery, Gli1 expression should be largely inhibited in normal keratinocytes and sensitive tumors cells. Therefore, Gli1 expression would not only suggest tumor identity, but also vismodegib resistance. We stained pre- and post-vismodegib sections from patients scored as "disease present" and "no sign of disease". While vismodegib treatment significantly *reduced* Gli1 expression (Fig 2A), we found that 82% of "no sign of disease" samples contained Gli1 positive micro-lesions (Fig 2A and 2B, Table 4). As expected, Gli1 expression was not significantly reduced in samples with "disease present" (Fig 2A), although two samples displayed no Gli1 staining (Table 4). Based on these data, we predicted that the micro-lesions present in post-treatment samples are likely residual tumor cells and may also be resistant to vismodegib.

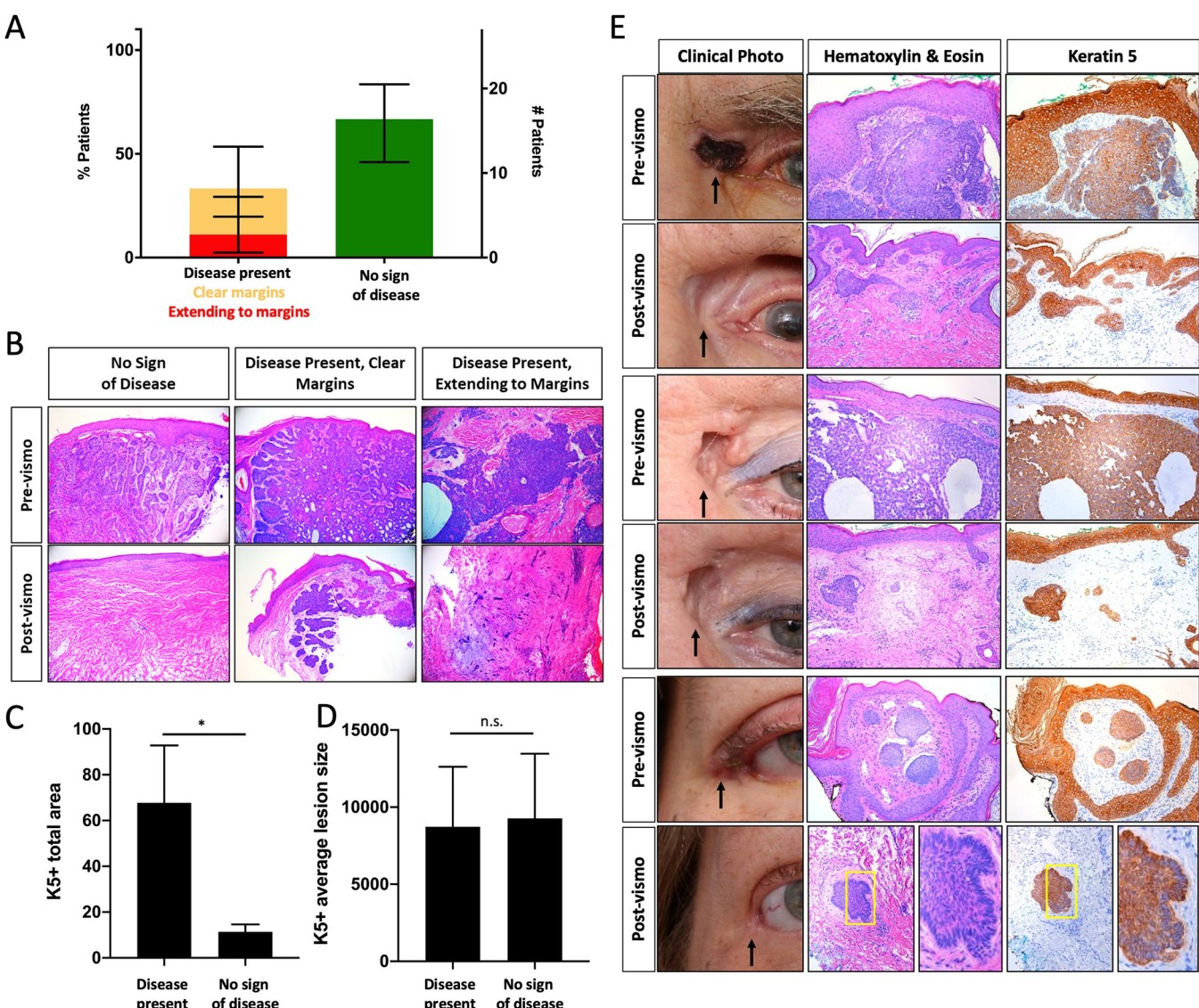

**Fig 1. Keratin positive micro-tumors present despite complete clinical response to vismodegib.** A) Pathology results for excision samples after vismodegib treatment (error bars indicate 95% CI, n = 9 (left column), n = 18 (right column)), B) Pre- and post-vismodegib histology, C) Average total area and D) Average lesion size (pixels) of keratin 5 positive staining (normalized to section length, error bars indicate SEM, unpaired t-test *p<0.05, n = 7 (left column), n = 11 (right column)), E) Clinical photo, histology, and keratin staining pre and post-vismodegib treatment of patients scored as "no sign of disease" (black arrow–tumor location, yellow box—area of peripheral palisading).

**Table 3. Keratin 5 expression in post-vismodegib excision samples.**

| No Sign of Disease | | |
| --- | --- | --- |
| | Average section length (pixels) | 5890 |
| | Average number of lesions per section | 10.3 |
| | Average lesion area (pixels) | 9273.2 |
| Disease Present | | |
| | Average section length (pixels) | 6743 |
| | Average number of lesions per section | 71.4 |
| | Average lesion area (pixels) | 8727.6 |

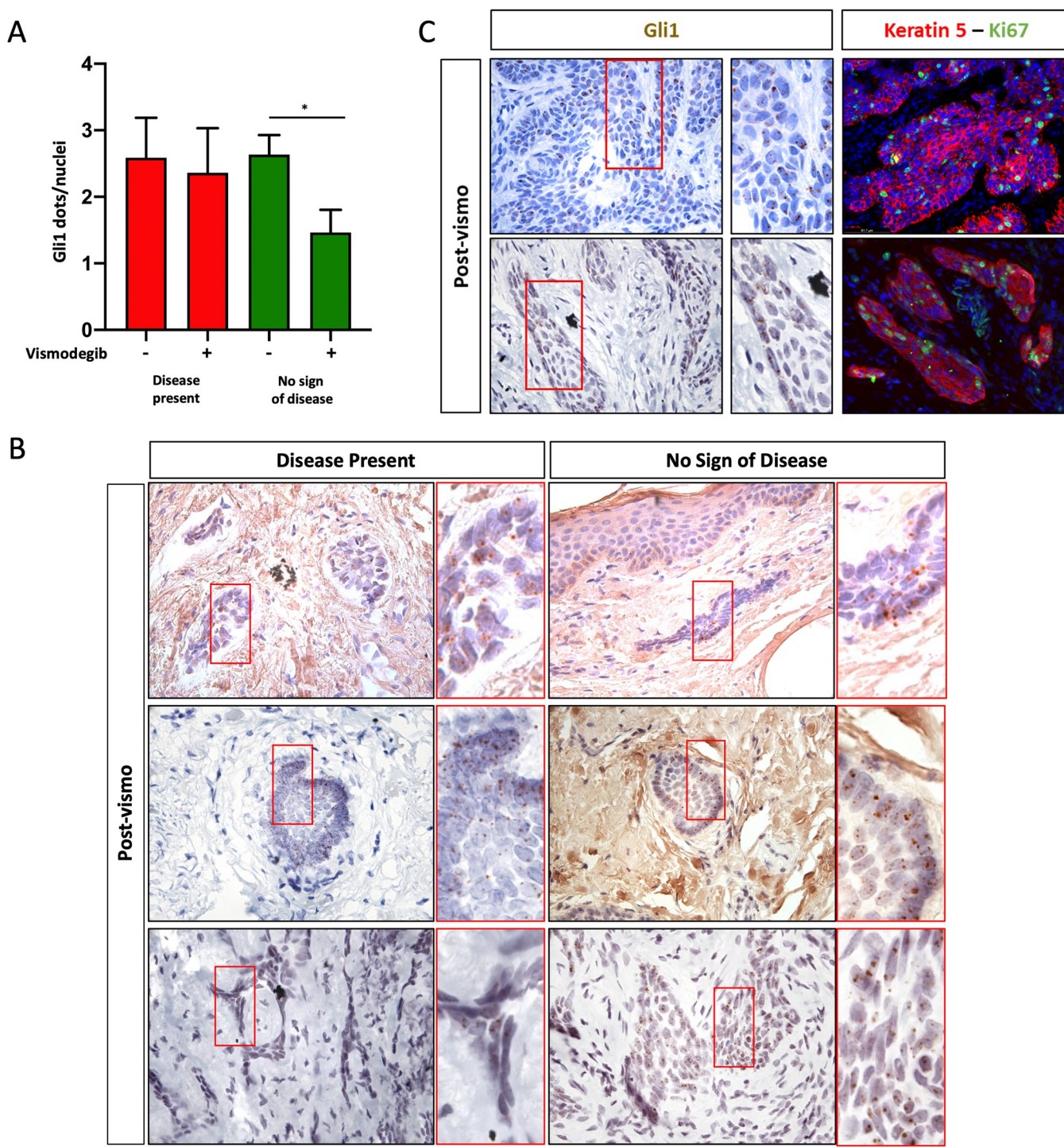

**Fig 2. Persistent BCC expresses Gli1 and is proliferative.** A) Gli1 expression (dots/nuclei) pre- and post-vismodegib treatment in patient samples scored as disease present (red) or no sign of disease (green) (error bars indicate SEM, unpaired t-test $^*$p<0.05, n = 7,6,9,11 columns left to right), B) Gli1 expression (brown dots) in excision samples from "disease present" (left panels) and "no sign of disease" (right panels) samples. C) Gli1 expression (left panels, brown) and proliferation (Ki67, green) in keratin 5 positive (red) persistent lesions from samples scored as disease present (top panels) and no sign of disease (bottom panels).

**Table 4. Gli1 expression in post-vismodegib excision samples.**

|  |  | # of subjects (%) |
| --- | --- | --- |
| No Sign of Disease |  | 11 (100) |
|  | Gli1 + | 9 (82) |
|  | Gli1 - | 2 (18) |
| Disease Present |  | 8 (100) |
|  | Gli1 + | 6 (75) |
|  | Gli1 - | 2 (25) |

By definition, resistant tumor cells display the ability to induce hedgehog signaling and proliferate in the presence of vismodegib. To test whether these Gli1 positive micro-lesions were proliferatively active, we stained consecutive sections from both "disease present" and "no sign of disease" samples and found that in both populations, lesions that display Gli1 expression also express K5 and the proliferation marker Ki67 (Fig 2C).

## Sensitive orbital BCC tumors harbor low frequency SMO variants

The ability of residual micro-lesions to express Gli1 and proliferate in the presence of vismodegib suggests they may harbor resistance mutations. Because the majority of primary BCC mutations are in Patched 1(PTCH1) and the majority of resistant mutations are in Smoothened (SMO) [9,10], we performed targeted exon sequencing of both PTCH1 and SMO on pre-treatment formalin-fixed paraffin-embedded (FFPE) biopsy sections to identify variants of interest in pre-vismodegib opBCC tumors. All samples displayed variants in either one or both genes (Fig 3A). In 19 biopsies from 17 patients, we detected eight known pathogenic PTCH1 variants and four pathogenic SMO variants (Fig 3B and 3C). The number of detected variants varied between patients. One biopsy harbored only 17 total variants, while one had 157 (Fig 3C). Most variants were single nucleotide substitutions (Fig 3D). We determined PTCH1 variants of interest by filtering for mutations present at an allele frequency of greater than 0.25, and further characterized each variant using the VarSome.com database (Fig 3E) [29]. Because we were interested in the possibility of low-frequency vismodegib-resistant variants, we characterized all non-silent SMO variants using VarSome. Four patients displayed previously characterized pathogenic SMO variants (Fig 3F). These included: V321M [30], L412F [30], A459V [31], and W535L [30] (Fig 3C).

## Vismodegib treatment selects for resistant tumor cells

Two years after study completion, one "no sign of disease" patient experienced a recurrence in the same location as the original tumor (Fig 4A). We performed Gli1 *in situ* hybridization on pre-, post-vismodegib, and the recurrent tumor samples and found similar Gli1 expression in all three samples (Fig 4A). Targeted sequencing revealed six PTCH1 variants of interest, and two SMO variants of interest in the pre-treatment tumor. Five PTCH1 variants (S733C, 98242925 A→C, 98229389 C→G, 98221861 T→C, and P1315L) were considered non-pathogenic and likely germ-line, as they remained unchanged despite tumor regression post-vismodegib treatment (Fig 4B). However, one PTCH1 variant, P1399S, was present at approximately 0.5 allele frequency in the pre-treatment and recurrent tumor, but not in the post-treatment excision sample (Fig 4B) suggesting tumor specificity and pathogenicity, although this variant has not yet been described in the literature. Interestingly, this patient also harbored the SMO W535L mutation at a 0.1 allele frequency in the pre-treatment sample. This mutation is perhaps the most well characterized resistance-conferring mutation in BCC, also known as the

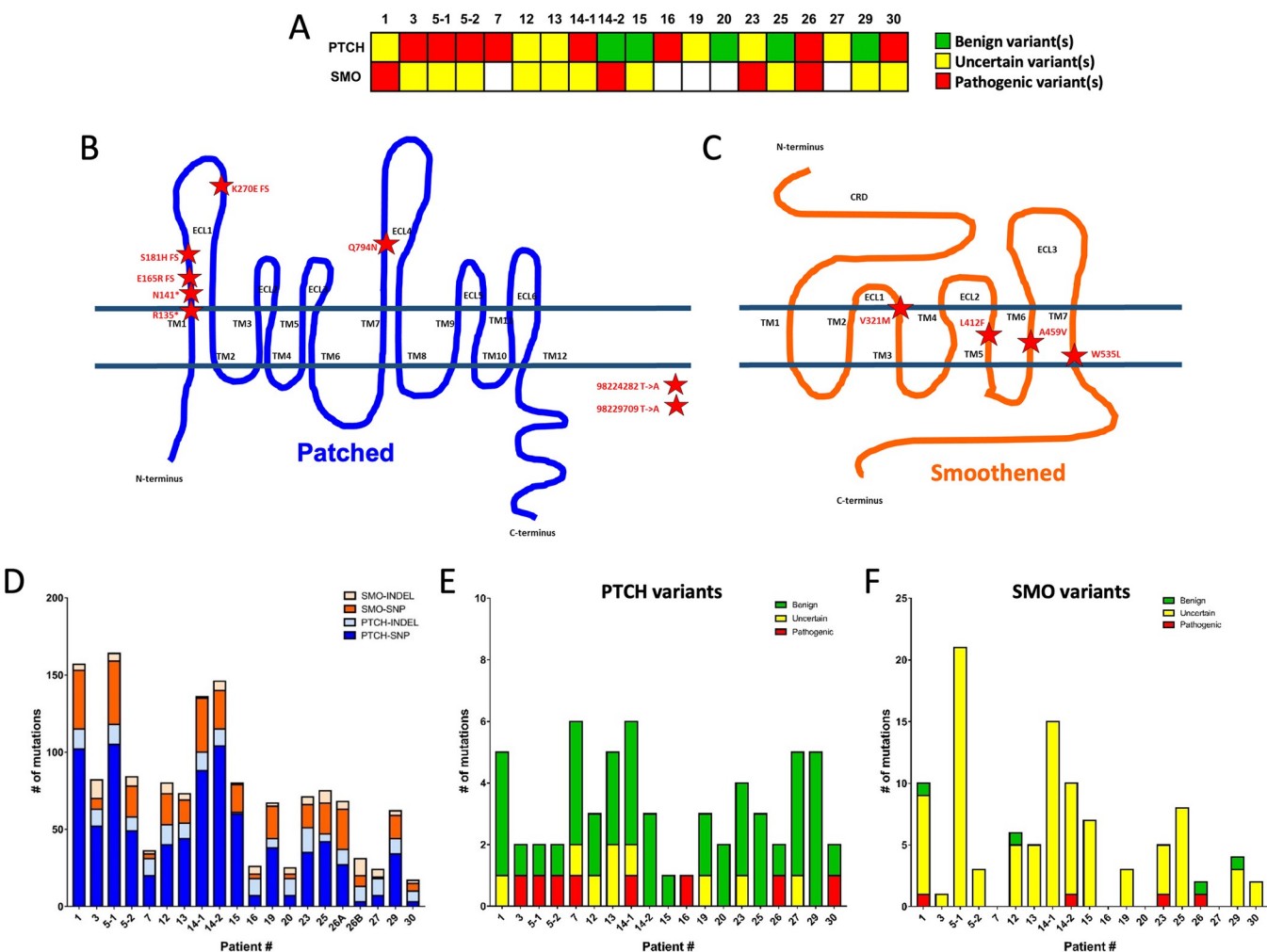

**Fig 3. Mutational status of pre-vismodegib orbital BCC tumors.** A) Summary of all PTCH and SMO variants present in each tumor. Schematic of PTCH (B) and SMO (C) pathogenic variants present in patient cohort. D) Summary of variant data by patient, E) Classification of non-silent PTCH variants with >0.25 allele frequency, F) Classification of non-silent SMO variants.

SMOM2 [2]. While this mutation was not detected in the post-treatment sample, the frequency was enriched in the recurrent tumor to 0.4, suggesting that vismodegib treatment selected for these cells, and the targeted sequencing assay was not sensitive enough to detect the micro-lesions present in the surgical sample (Fig 4B).

## Discussion

Prospective clinical trials form the foundation of evidence-based medical practice. Both the visual function and tumor response outcomes from the VISORB clinical trial highlight the utility of vismodegib for advanced opBCC [25]. Vismodegib as a monotherapy and as a neoadjuvant is clearly effective in reducing tumor burden and preserving visual organs. However, further characterization of samples from vismodegib-treated patients has uncovered a need for extreme caution when determining whether patients display a "complete response." Historically, patients' response to vismodegib is measured by either physical measurements and/or pathologic evaluation of biopsies/excision samples post-treatment. In the pivotal study of

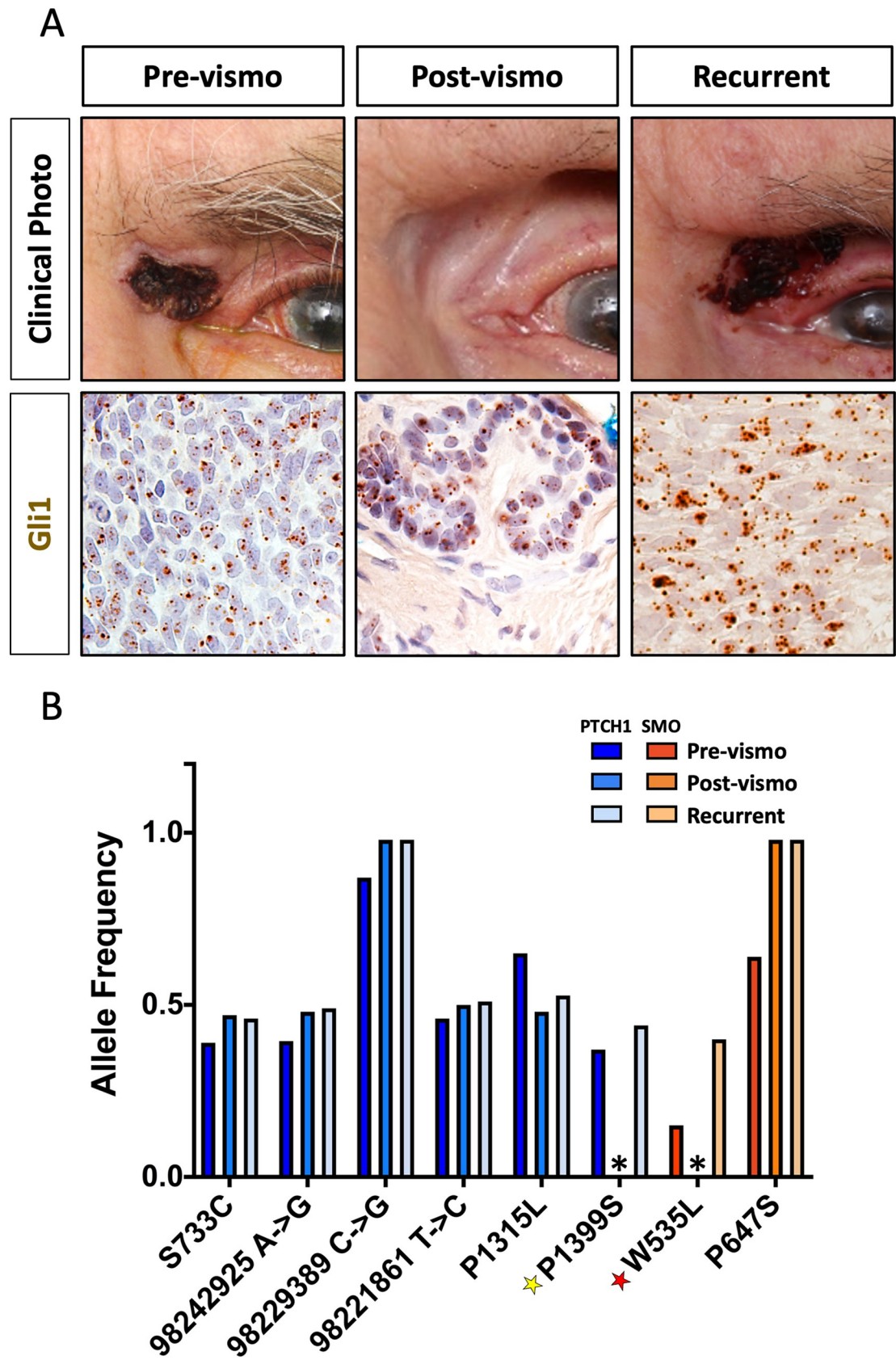

**Fig 4. Vismodegib treatment enriches for cells with resistant mutations.** A) Clinical photos and Gli1 expression (brown dots) in a patient pre- (left panels), post-vismodegib (center panels), and after recurrence (right panels), B) Allele frequency of PTCH1 and SMO variants pre-, post-vismodegib, and after recurrence (*variant not detected)(yellow star–predicted pathogenic variant, red star–known pathogenic variant).

vismodegib, patients underwent biopsy at the time of best response, where one to five, 3 mm punch biopsies or core needle biopsies were taken from the original tumor site and evaluated [4]. 34 of 63 (54%) patients displayed no residual disease in biopsy samples [4]. This response rate is similar to our own study, in which 67% of patients who underwent excision showed "no sign of disease" based on pathologic evaluation. However, after performing Gli1 *in situ* hybridization on these samples, we found that 82% of "no sign of disease" samples contained Gli1 positive micro-lesions (Fig 2, Table 4). While conventional histology is adequate for margin control of vismodegib-naïve tumors, our results suggest that this method can miss residual and potentially resistant cells.

Understanding the mutational status of an advanced BCC tumor provides insight into the best course of treatment for patients. Here we show the ability of targeted sequencing on FFPE pre-vismodegib biopsy samples to detect potential high-frequency primary mutations and low-frequency resistance-conferring mutations in pre-treatment tumors. Despite the presence of residual tumor cells in both "no sign of disease" and "disease present" samples, we were unable to detect pathogenic mutations in post-vismodegib surgical samples from either group (Fig 4). Because sequencing was performed using whole excision sample FFPE sections, we believe the mutational analysis assay is simply not sensitive enough to detect residual micro-lesions in post-treatment samples due to the relatively small number of residual tumor cells versus surrounding normal keratinocytes and dermal cells.

The data we have gathered from the VISORB clinical trial and this associated study provide new insight into how opBCC tumors respond to vismodegib. In previous trials and VISORB, vismodegib treatment appears to provide complete tumor clearance in many patients based on physical measurements and traditional pathology [4–8,25]. However, here we provide practice-changing evidence of potentially resistant residual disease in patients displaying complete response. This residual disease is visible in H&E stained sections, but more apparent in Keratin 5 and Gli1 *in situ* stained sections. In addition, our data highlight the ability of pre-vismodegib targeted sequencing to identify low-frequency resistance mutations which may increase the potential for recurrence in patients. Importantly, most occult residual disease appeared in dermal specimens, sparing deep orbital tissue; this suggests that tumor elements invading the orbit responded preferentially to hedgehog inhibition, providing a key justification for using neo-adjuvant hedgehog inhibition in orbit-involving tumors and likely other tumors that invade in 3-dimentional body spaces.

The majority of prospective clinical trials are focused on clinical outcomes and their potential associations with molecular markers. Here we show that a basic molecular and genetic analyses of patient tissue samples can significantly alter the interpretation of such outcomes. Specifically, this study reveals that standard approaches for margin control are less predictive for vismodegib-treated tissues, a conclusion that may extend to other trials and experimental conditions. Furthermore, many clinical trials require expensive long-term follow-up, e.g. treatment of basal cell carcinoma in which recurrences may take years to manifest [10,16]. Adding molecular and genetic analyses may allow for improved outcome analysis in a shorter amount of time, while providing molecular insights into the mechanisms of response or lack thereof. Lastly, while vismodegib is commonly used as a long-term therapy for advanced BCC, our data suggest that its best use may be as a neoadjuvant, maximizing chemoreduction (particularly of deep margins) while minimizing selection of resistant alleles prior to definitive excision

in which margins are assessed by more than routine H&E staining (with additional sections and/or histologic markers). The VISORB clinical trial and this follow up molecular analysis represent a unique opportunity to assess the utility of prospectively linking clinical, molecular and genetic studies in order to reach the correct conclusions in the shortest possible time.

## Materials and methods

### Experimental model and subject details

**Human studies.**   opBCC samples were obtained as a part of the VISORB clinical trial with written informed consent from each patient as approved by the Institutional Review Board (IRBMED) of the University of Michigan Rogel Cancer Center (IRB approval # HUM00082579; Clinical trial NCT02436408).

### Method details

**Immunohistochemistry.**   FFPE sections were utilized for all staining experiments. For immunofluorescence experiments the following antibodies were used: guinea pig anti-cytokeratin 5 (Abcam ab194135, 1:1000), rat anti-Ki67 (Invitrogen SolA15, 1:500).

**Targeted sequencing.**   DNA was isolated from FFPE pre-vismodegib biopsies and surgical excision samples using the QIAamp DNA FFPE Tissue Kit (Qiagen). Libraries were prepared using a custom Accel-Amplicon Custom Panel (Swift Biosciences), variant calling was performed by Swift Biosciences. Filter criteria for variant calling by Swift included: minimum coverage of 10 reads, strand bias multiple correction false discovery rate (FDR) corrected p-value >0.001000, and minimum Phred score of 58 for SNVs and 37 for INDELS. Non-silent PTCH1 variants present at an allele frequency of 0.25 or greater and all non-silent SMO variants were further characterized using VarSome.com.

**Statistics.**   All data are presented as means ± standard error of mean. Unpaired *t*-tests were performed using GraphPad Prism 8, and results from these tests are indicated in the figures and legends.

## Supporting information

**S1 Graphical abstract.**
(TIF)

**S1 File. Targeted sequencing files 1–25.**
(ZIP)

## Author Contributions

**Conceptualization:** Alon Kahana.

**Data curation:** Christina F. Tingle, Curtis J. Heisel, Emily A. Eton, May P. Chan, Scott C. Bresler, Alon Kahana.

**Formal analysis:** Shelby P. Unsworth, Curtis J. Heisel, Emily A. Eton, Christopher A. Andrews, May P. Chan, Scott C. Bresler, Alon Kahana.

**Funding acquisition:** Alon Kahana.

**Investigation:** Shelby P. Unsworth, Christina F. Tingle, Emily A. Eton, Scott C. Bresler, Alon Kahana.

**Methodology:** Shelby P. Unsworth, Christina F. Tingle, Curtis J. Heisel, Christopher A. Andrews, May P. Chan, Alon Kahana.

**Project administration:** Alon Kahana.

**Resources:** Shelby P. Unsworth, Alon Kahana.

**Software:** Christopher A. Andrews.

**Supervision:** Alon Kahana.

**Validation:** Shelby P. Unsworth, Christopher A. Andrews.

**Writing – original draft:** Shelby P. Unsworth, Christopher A. Andrews, Alon Kahana.

**Writing – review & editing:** Shelby P. Unsworth, May P. Chan, Scott C. Bresler, Alon Kahana.

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
