## [Decision Letter · Decision Letter 0]

1 Sep 2022

PONE-D-22-05819

Analysis of Residual Disease in Periocular Basal Cell Carcinoma Following Hedgehog Pathway Inhibition: Follow Up to the VISORB Trial.

PLOS ONE

Dear Dr. Kahana,

Thank you for submitting your manuscript to PLOS ONE. After careful consideration, we feel that it has merit but does not fully meet PLOS ONE’s publication criteria as it currently stands. Therefore, we invite you to submit a revised version of the manuscript that addresses the points raised during the review process.

Your manuscript describing molecular characterization of residual BCCs following the VISORB trial could be of significant interest to the field, particularly in understanding tumor response to currently available Hh-targeted therapy in patients with advanced BCCs. However, concerns were raised regarding the technical aspects related to the mutational status and burden. Please address Reviewer 2’ concerns about Figs 3 and 4 and include the details about sequencing and filtering criteria and subsequent statistic analysis.

We look forward to receiving your revised manuscript.

Kind regards,

Arianna L. Kim, PhD

Academic Editor

PLOS ONE

Journal Requirements:

“This study was funded by grants to AK from Genentech, Inc., the University of Michigan Rogel Comprehensive Cancer Center, the Department of Ophthalmology and Visual Sciences, Research to Prevent Blindness, Inc., and a Head and Neck SPORE award from the National Cancer Institute”

“AK: Research Grants from Genentech, Research to Prevent Blindness, Kellogg Eye Center, and The University of Michigan Rogel Comprehensive Cancer Center.

“AK had served briefly as a consultant for Genentech/Roche for the purpose of discussions with the FDA. There are no other competing interests by AK or any of the other authors.”

Reviewers' comments:

Reviewer's Responses to Questions

**Comments to the Author**

1. Is the manuscript technically sound, and do the data support the conclusions?

Reviewer #1: Yes

Reviewer #2: No

2. Has the statistical analysis been performed appropriately and rigorously? 

Reviewer #1: Yes

Reviewer #2: I Don't Know

3. Have the authors made all data underlying the findings in their manuscript fully available?

Reviewer #1: Yes

Reviewer #2: No

4. Is the manuscript presented in an intelligible fashion and written in standard English?

Reviewer #1: Yes

Reviewer #2: Yes

5. Review Comments to the Author

Reviewer #1: I think the paper is innovative and brings light to procedures and investigations that should be performed also in other advance BCC locations.

It was a pleasure to review this paper.

The Gli-expression in residual tumor should be performed even in advance BCC localized in other than periocular tissues. It is crucial to understand the biology of these particular tumors and the reason some of these patients do not respond in the area of new treatment and advancing new protocols such as neo adjuvant treatment with HH inhibitors.

Please add in the introduction after reference 10 "While vismodegib has a relatively high response rate, unfortunately a subset of these tumors become resistant and recur" [9, 10 ) following reference:

Dika E, Scarfì F, Ferracin M, Broseghini E, Marcelli E, Bortolani B, Campione E, Riefolo M, Ricci C, Lambertini M. Basal Cell Carcinoma: A Comprehensive Review. Int J Mol Sci. 2020 Aug 4;21(15):5572. doi: 10.3390/ijms21155572. PMID: 32759706; PMCID: PMC7432343.

The results show one of the first studies that describe pathogenic variants of PTCH1, and SMO. The authors further characterized each variant using the VarSome.com database. This kind of multi center studies offers to the literature and most importantly to clinicians a better knowledge and an inspiration for further investigations.

Reviewer #2: Unsworth et al. present a study of residual disease in periocular (op) BCC following Hedgehog Pathway Inhibition.

The authors revealed that in treated with vismodegib opBCC patients who demonstrated a complete response histologically, excision samples contained micro-tumors.

In one case of recurrent tumor a SMO W535L mutation was identified. Sequencing of pretreatment tumor, also revealed a SMO W53L mutation but with a low frequency.

Major comment

The number of detected mutations is not realistic. The authors can consult other papers on BCC, such as Bonilla et al 2018, NatGen, to assess the average number PTCH1 or SMO mutations per tumor.

It seems that listed mutations include technical sequencing artifacts (Fig.3) and the germline variants (Fig 4B).

It is also possible from fig 4B that in the post-treatment tumor sample no tumor cells were present.

The methods should include details about the sequencing and filtering criteria for the variants.

6. PLOS authors have the option to publish the peer review history of their article (what does this mean?). If published, this will include your full peer review and any attached files.

Reviewer #1: **Yes: **Emi DIKA

Reviewer #2: No

---

## [Author Response · Author response to Decision Letter 0]

17 Oct 2022

We thank the reviewers for their comments and suggestions. All the suggested references have been added.

Reviewer #2 is correct that the average number of PTCH1 or SMO “mutations” expected to be found in a basal cell tumor is low; however, in this manuscript we are not reporting tumor-specific mutations but instead all variants present in the whole FFPE tumor biopsy or post-treatment excision sample. These samples include tumor cells, normal keratinocytes, and surrounding dermal cells, meaning our sequencing results include both germ-line variants and tumor-specific mutations. Because we do not have matched non-tumor control skin for study patients – we are unable to distinguish between germ-line and tumor specific mutations. Instead, we characterized all non-silent PTCH variants with a greater than 0.25 allele frequency and all non-silent SMO variants using VarSome to identify which were known to be pathogenic and thus more likely to be tumor-specific. To make this more apparent in the manuscript, we removed all references to “mutations” which suggest tumor specificity, and instead referred only to “variants of interest” present in the samples. We also clarified that sequencing was performed on “pre-treatment formalin-fixed paraffin-embedded (FFPE) biopsy sections”

As the reviewer stated – while it is possible that no tumor cells were present in the post-treatment sample for the patient that recurred, the presence of Gli1+ proliferative cell clusters throughout the excision sample suggest otherwise. Instead, we believe that because the excision samples contain a relatively small number of these residual tumor cells compared to surrounding normal keratinocytes and dermal cells the residual tumor cells were undetectable using our targeted sequencing methods. It is possible that these cells could be isolated using laser micro-dissection followed by targeted sequencing; however, this was outside the scope of our study. This has been further described in paragraph 2 of the discussion.

We are grateful for the helpful comments to improve our manuscript.

Sincerely,

Alon Kahana and co-authors

---

## [Editor Report · Decision Letter 1]

1 Nov 2022

Analysis of Residual Disease in Periocular Basal Cell Carcinoma Following Hedgehog Pathway Inhibition: Follow Up to the VISORB Trial.

PONE-D-22-05819R1

Dear Dr. Kahana,

We’re pleased to inform you that your manuscript has been judged scientifically suitable for publication and will be formally accepted for publication once it meets all outstanding technical requirements.

Kind regards,

Arianna L. Kim, PhD

Academic Editor

PLOS ONE
---

## [Editor Report · Acceptance letter]

23 Nov 2022

PONE-D-22-05819R1 

Analysis of Residual Disease in Periocular Basal Cell Carcinoma Following Hedgehog Pathway Inhibition: Follow Up to the VISORB Trial. 

Dear Dr. Kahana:

I'm pleased to inform you that your manuscript has been deemed suitable for publication in PLOS ONE. Congratulations! Your manuscript is now with our production department. 

Kind regards, 

on behalf of

Dr. Arianna L. Kim 

Academic Editor

PLOS ONE